# Introducing the EMPIRE Index: A novel, value-based metric framework to measure the impact of medical publications

**Avishek Pal** [1], **Tomas James Rees** [2] *

**1** Novartis Pharma AG, Basel, Switzerland, **2** Oxford PharmaGenesis, Oxford, United Kingdom

☯ These authors contributed equally to this work.
\* tomas.rees@pharmagenesis.com

**Data Availability Statement:** All data files used in the analysis and processed result used to create the figures are available on Figshare: Rees, Tomas (2021): EMPIRE Index development data. figshare.

## Abstract

Article-level measures of publication impact (alternative metrics or altmetrics) can help authors and other stakeholders assess engagement with their research and the success of their communication efforts. The wide variety of altmetrics can make interpretation and comparative assessment difficult; available summary tools are either narrowly focused or do not reflect the differing values of metrics from a stakeholder perspective. We created the EMPIRE (EMpirical Publication Impact and Reach Evaluation) Index, a value-based, multi-component metric framework for medical publications. Metric weighting and grouping were informed by a statistical analysis of 2891 Phase III clinical trial publications and by a panel of stakeholders who provided value assessments. The EMPIRE Index comprises three component scores (social, scholarly, and societal impact), each incorporating related altmetrics indicating a different aspect of engagement with the publication. These are averaged to provide a total impact score and benchmarked so that a score of 100 equals the mean scores of Phase III clinical trial publications in the *New England Journal of Medicine* (NEJM) in 2016. Predictor metrics are defined to estimate likely long-term impact. The social impact component correlated strongly with the Altmetric Attention Score and the scholarly impact component correlated modestly with CiteScore, with the societal impact component providing unique insights. Analysis of fresh metrics collected 1 year after the initial dataset, including an independent sample, showed that scholarly and societal impact scores continued to increase, whereas social impact scores did not. Analysis of NEJM 'notable articles' showed that observational studies had the highest total impact and component scores, except for societal impact, for which surgical studies had the highest score. The EMPIRE Index provides a richer assessment of publication value than standalone traditional and alternative metrics and may enable medical researchers to assess the impact of publications easily and to understand what characterizes impactful research.

Dataset. https://doi.org/10.6084/m9.figshare.
14256305.

**Funding:** Research and editorial support
(manuscript proofreading, figure drawing, and
project management) was funded by Novartis
Pharma AG (https://www.novartis.com) and
conducted by Oxford PharmaGenesis, who jointly
planned the study, collected and analysed the data,
made the decision to publish and prepared the
manuscript.

**Competing interests:** I have read the journal's
policy and the authors of this manuscript have the
following competing interests: Avishek Pal is an
employee of Novartis Pharma AG. Tomas Rees is
an employee of Oxford PharmaGenesis and
received funding for research and editorial support
(manuscript proofreading, figure drawing, and
project management) from Novartis Pharma AG for
this study. This does not alter our adherence to
PLOS ONE policies on sharing data and materials.

## Introduction

The publication of clinical trial results and other medical advances is an ethical obligation and benefits a variety of stakeholders. Published information can be used by physicians, other healthcare practitioners, and patients to evaluate and understand potential treatments. Medical researchers and academics can use published results to inform their own research endeavors and to advance medical research. In addition, policymakers use published information to develop guidelines and treatment protocols that help to guide changes to clinical practice.

Publications are therefore vehicles for communicating research insights for peer-to-peer validation and discussion. Article-level metrics provide an indication of the reach, engagement, and impact of publications, but they cannot be assumed to provide a measure of the quality (or even the full impact) of the underlying research. One study found altmetric scores h to be highly correlated with expert assessment of research impact, but not correlated with assessment of research quality [1], while another found no correlation between altmetrics scores or citations and the impact of research as assessed in the UK's Research Excellence Framework [2]. Publication metrics can contribute to research assessment if conducted within a comprehensive framework that also assesses non-publication impact, such as the Becker Model [3].

Impact measurements aim to assess the utility of published research for its intended audience as well as the effectiveness of the communication. Objective measures of impact can support these endeavors by enabling comparative assessments to be made. However, making such measurements is challenging owing to the lack of available data and agreed definitions of impact. Historically, a common proxy for the publication impact of an article has been the impact factor of the journal in which it is published. However, although the journal impact factor (JIF) may help to identify journals with a high readership, it is widely recognized to be a poor indictor of the quality or impact of individual research articles [4, 5].

Article-level metrics avoid the category error of using JIF in this context. The number of citations is the most well-known metric, but this reflects only scholarly activity and citations can take years to accumulate [6]. Recently, the advent of alternative article-level metrics (altmetrics) has provided a new way to evaluate the impact of scientific publications. A wide range of potential altmetrics exists, signifying different interactions with the publication of interest but differing widely in quality and representativeness [7]. The sheer volume of potential metrics is evident in the information gathered by major aggregators including Altmetric, which collects nearly 20 different altmetrics, and PlumX, which collects over 40 [8, 9].

To make metrics easier to interpret, various approaches have been taken to distilling them into simplified scores. The most well-known of these is the Altmetric Attention Score (AAS), which weights a variety of individual metrics to reflect a subjective assessment of relative reach and aggregates them into a single number. Attempts to reduce any complex set of metrics into one linear scale have been criticized because they will tend to be driven by a single predictor, especially when the variables included are correlated [10]. Indeed, the AAS is dominated by Twitter and, to a lesser extent, news articles [10–12], so it does not reflect the impact of publications among researchers or policy-makers. Furthermore, the AAS has been criticized for arbitrary (and at times opaque) weighting of components [13, 14].

The full range of altmetrics is, however, multifactorial because they have diverse origins and represent different activities relating to publications [15–19]. The AAS is only weakly correlated with the number of citations [20, 21]. Among the most cited, downloaded, and mentioned articles published in general medical journals, only 2.5% were found in all three lists [22]. This implies that altmetrics cannot effectively be reduced to a single linear representation, and data reduction can, at best, provide several scores that group together related metrics. As a

result, a metric system with summary scores designed on data-reduction principles must, if it includes diverse, weakly correlated metrics, provide for several distinct factors [19, 23].

We sought to develop a value-based, multi-component metric framework for medical publications, the EMPIRE (EMpirical Publication Impact and Reach Evaluation) Index, that would allow authors and other professionals within the medical and pharmaceutical fields to assess the impact of publications in terms meaningful to them. The metric framework is also intended to monitor the long-term impact of publications, predict the likely long-term impact using early indicators, and identify the effectiveness of communication efforts surrounding publications.

Focusing on a single discipline, medicine, has several advantages when developing a metric framework. First, value is inherently subjective and is likely to differ between disciplines. Similarly, the relationship between metrics varies between scientific disciplines [15, 20, 24, 25]. Second, using the number of citations alone is known to underestimate severely the impact of clinical intervention research compared with basic and diagnostic medical research [26], underscoring the need for a multivalent approach to impact assessment. Third, medicine and medical sciences is the scientific discipline richest in metrics [20], providing a large dataset to examine.

## Materials and methods

### Approach to developing the scoring system

Development of the scoring system for the EMPIRE Index proceeded through a series of stages, outlined in Fig 1 and described in more detail in the sections below.

In summary, during **framework construction**, a large set of publications was generated to gain an in-depth understanding of the statistical characteristics of altmetrics in a relevant sample. Publications of Phase III clinical trials were chosen for analysis because these studies typically require a high investment of resources and personnel and are most likely to have an impact on clinical practice. In addition, they are likely to be rich in metrics–the mean number of metric counts has a substantial effect on the size of the intercorrelation observed in a publication sample [27]. A series of statistical analyses was then conducted to determine which metrics were comprehensive and provided useful information, and how they were related to each other. The grouping and **weighting** of metrics was informed by these analyses but was ultimately driven by an understanding of the type of interaction each metric represented and by value judgments provided by a panel of stakeholders.

Once the structure and weighting of the metric system had been decided, **predictor scores** were developed using altmetrics that accumulated rapidly. Scores for all components of the system were then scaled to a **benchmark** representing a very high level of impact; for this, Phase III articles published in the *New England Journal of Medicine* (NEJM) were chosen.

The last stage in development was to **characterize** the performance of the final scoring system. This was carried out in three datasets: the original Phase III dataset, the Phase III dataset with metrics updated after 1 year (and including 1 new year's worth of publications), and a dataset comprising publications selected by NEJM editors that were likely to influence clinical practice.

### Sample acquisition

**Reference Phase III sample.** We identified a sample of publications (the reference Phase III sample) that was representative of the primary output of clinical medicine (Phase III clinical trials) as well as being sufficiently large to permit statistical and longitudinal analysis. Data were obtained across 3 years of publications to ensure the sample was large enough for analysis

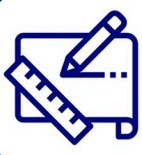

### Framework construction
- A reference sample of Phase III clinical trial publications was analyzed to determine statistical underpinnings
- Related metrics were grouped based on this analysis and end-user utility

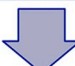

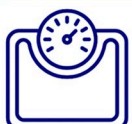

### Weighting
- Metrics were weighted based on stakeholder insights into value, taking into account the prevalence in the reference sample

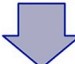

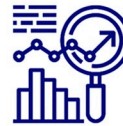

### Predictor scores
- Early-onset metrics were used to create a prediction scale for likely long-term scores

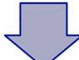

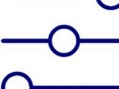

### Benchmarking
- A benchmark target was selected (average Phase III trials in the NEJM = 100 score) and scores scaled accordingly

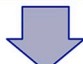

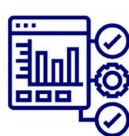

### Characterization
- The performance of the framework was assessed:
  1. In samples of Phase III studies used to develop the score and in prospectively collected samples
  2. In a sample of 'practice changing' articles collected by the NEJM

**Fig 1. Process for developing the scoring system.** NEJM, *New England Journal of Medicine.*

and included publications old enough to have accumulated citations in guidelines and policy documents, while minimizing the impact of confounding factors related to the change in use of publications over time (in particular, changes in social media mentions). Non-English publications were excluded because the distribution of altmetrics for these was likely to differ substantially from that of publications in English (e.g. news coverage). The search was conducted on May 23, 2019 in PubMed, using the search term: ("clinical trial, phase iii"[Publication Type]) AND (("2016/05/01"[Date—Publication]: "2019/05/01"[Date—Publication]) AND Clinical Trial[ptyp] AND English[lang]).

Altmetrics for this sample were obtained on May 27, 2019. Article publication dates were obtained from Altmetric Explorer and were used to split the sample into two subsamples–the

older 50% (1H) and the younger 50% (2H)–to assess the effect of temporal change in altmetrics.

**Benchmark NEJM Phase III sample.**   The benchmark sample provided a 'target' against which to calibrate metrics achieved by other publications. For this reason, a sample was chosen from a journal widely considered the 'gold standard' for clinical trial publications, the NEJM, which has the highest JIF of all general medical journals and describes itself as "the world's leading medical journal and website" [28]. The benchmark sample comprised all Phase III clinical trial articles published in the NEJM in 2016 (manually identified from a sample of all clinical trials obtained via a PubMed search). The year was selected to allow the accumulation of metrics such as article or guideline citations, and to match the base year in the reference Phase III sample. Altmetrics for the benchmark sample were obtained on July 31, 2019.

**1-year update Phase III sample.**   An independent sample was obtained to assess the metric framework for consistency. This sample was identified on June 6, 2020 using the same search terms as the reference Phase III sample but for the consecutive 12-month period (i.e. ("clinical trial, phase iii"[Publication Type]) AND (("2019/05/01"[Date—Publication]: "2020/05/01"[Date—Publication]) AND Clinical Trial[ptyp] AND English[lang])). Metrics for this 1-year update Phase III sample as well as for the original reference Phase III sample were acquired on June 7, 2020 (approximately 1 year after the original metrics were acquired).

To enable analysis of temporal changes, both the updated reference sample and the prospective Phase III sample were divided into 12-month subsamples (May 1 to April 31) based on publication dates provided by Altmetric Explorer. Publications with a publication date before May 1, 2016 according to Altmetric Explorer were excluded.

**NEJM notable articles sample.**   An additional independent sample was identified with which to assess framework performance in other types of clinical research, especially the utility of the societal impact component. Annually, the editor of the NEJM curates a selection of articles published in the journal that year that they believe have practice-changing potential ('notable articles'). We identified all of these articles for the years 2016, 2017, 2018, and 2019 [29–32], and obtained altmetrics for them on January 8, 2020. Articles were classified by the authors under a broad typology: interventional (studies describing an intervention with a medical treatment intended for clinical practice), observational (prospective and retrospective non-interventional studies), innovative (publications describing novel techniques or assays), and surgical.

### Acquisition of altmetrics and other metrics

Data for all publications were obtained from the five sources listed below.

- Altmetric Explorer [9]: This was the primary source for altmetrics data as well as publication dates).

- PlumX [8]: In addition to a wide range of metrics similar to those provided by Altmetric Explorer, PlumX provided some unique metrics such as citations in articles classified by Medline's indexers as 'clinical practice guideline' (PubMed guidelines).

- Pubstrat Journal Database [33]: This was scraped to determine JIFs for journals identified by Altmetric Explorer in the acquired datasets.

- CiteScore [34]: A journal-level, citation-based metric, similar to JIF. CiteScore was downloaded for all journals on August 7, 2019 and CiteScore values for 2016 were used.

- Scimago Journal Ranking [35]: A journal-level, citation-based metric that used a PageRank algorithm.

In addition to these standard metrics, original tweets and retweets (provided by Altmetric. com) were obtained for the reference Phase III sample.

In a similar way to the exploratory analysis of Costas et al. (2015), an 'altmetrics-driven' universe of publications was created in which all publications had at least one altmetric or citation (via Altmetric Explorer) [17]. Costas et al. noted that this analysis did not result in a meaningful impact on the precision of altmetrics as predictive tools for citations, but did reduce the zero inflation that can confound statistical analysis.

## Statistical analysis

Analyses were conducted in Microsoft Excel using the Analyse-it plugin (Analyse-it Software, Ltd., Leeds, United Kingdom). Descriptive statistics were obtained and Spearman rank correlations between individual altmetrics were calculated. In addition, exploratory factor analysis was used to provide insights into how best to group similar metrics. Factor analysis assumes that latent or underlying factors exist that causally influence the observations. For the purposes of metric development, we wanted to explore the hypothesis that publications have an intrinsic 'social' interest leading to social media mentions that is fundamentally different from an intrinsic 'scholarly' interest leading to citations. An alternative data-reduction technique, principal component analysis, simply creates one or more index variables explaining as much statistical variance as possible without regard to theoretical differences in the metrics. In practice, the two approaches yield similar results.

We used maximum likelihood factor analysis with oblique (oblimin) rotation. Because altmetrics follow a power-law distribution [36], data were log-transformed before factor analysis. All data were increased by 1, which allows the discretized lognormal distribution to be fitted to the full range of data [37]. Adding a positive constant to the dependent variable is a common solution to the problem of log-transformation of datasets containing zeros, although it does introduce a small distortion to the data [38]. Regression analyses were conducted using multiple linear regression on the untransformed data.

EMPIRE Index scores for the NEJM notable articles were averaged over the different years (2016–2019). To control for the impact of time on the accumulation of altmetrics, EMPIRE Index scores for articles in each year were expressed as a percentage of the average score of observational studies (the highest-scoring article type), and the average of these yearly percentages was taken.

## Value assessment

An internal Novartis cross-functional stakeholder panel meeting was convened on July 9, 2019, comprising representatives from scientific communications, medical, commercial, launch strategy, and medical analytics departments. Participants reviewed information on the analyses conducted as well as background information on metrics, and provided qualitative insights into the interpretation and importance of key metrics. Quantitative value assessments were obtained through points allocation (i.e. participants were given a 7 points to allocate among the 11 metrics according to the ones they felt best represented social impact, and a further 7 each to allocate according to which metrics they felt represent scholarly and societal impact (i.e. 21 points in total). Voting was conducted openly and in a single round. Points were summed for each metric and the proportion of points allocated to each metric was calculated.

## Predictor scores

Two predictor scores were developed based on metrics that accumulate rapidly after publication. The early predictor score included altmetrics that accumulated most rapidly (Twitter,

Facebook, and news mentions) [6, 39, 40], and included CiteScore, used here as a proxy for the readership and interest in a journal. The intermediate predictor score included blog mentions, F1000Prime mentions, and Mendeley readers–altmetrics that accumulate more slowly, but still faster than metrics with high lag, such as citations.

The basis of each predictor score was a multiple linear regression of the altmetrics included in the predictor against the total impact score in the reference Phase III sample. Weightings for each metric were calculated as follows:

$$weighting = \beta \frac{sum_m}{\int sum_{m1,m2...}}$$

where β is β from linear regression, $sum_m$ is the sum total of the incidence of the target metric in the reference sample, and $sum_{m1,m2...}$ is the sum total of all metrics included in the predictor score.

## Case examples

A total of 59 publications including Novartis-sponsored research were tracked over 9 months in three quarterly reports from May-December 2020. From these assessments publications with notable metrics were identified for further analysis. Two case examples of interest are reported here.

## Results

### Framework construction

The initial search found 3498 Phase III clinical publications, of which altmetrics for 3450 were identifiable by PlumX and 2891 by Altmetric Explorer. The analysis set comprised 2891 articles with at least one metric identified by Altmetric Explorer, of which eight were unavailable in the PlumX dataset. Publication metric characteristics of this sample are shown in S1 Table. Several altmetrics had a very low density so were discarded for further analysis (e.g. Weibo, LinkedIn, Google+, Pinterest, Q&A, peer review, video, and syllabi mentions). Some altmetrics were retained despite a low density as they were thought to provide unique insights relevant to the objectives (policy, patent, F1000Prime, Wikipedia, and guideline [from PlumX] mentions). Some metrics of high relevance (abstract and publication views and downloads) were discarded because the quality of the data was inconsistent–in particular, many papers had numerous citations and Mendeley readers without recorded views or downloads, suggesting that coverage was incomplete.

Journal-level metrics were not included in the EMPIRE Index total impact score or component scores, but they were considered potential components of predictor scores. Given that the coverage obtained with CiteScore was higher than with the other two journal-level metrics examined (JIF and Scimago Journal Ranking–S1 Table part C), CiteScore was selected for further analyses.

Pairwise Spearman correlations between the altmetrics included are shown in S2 Table. The most common metrics were strongly correlated (news, blog, and Twitter mentions, Mendeley readers, and Dimensions citations). The strongest correlation was seen between Mendeley readers and Dimensions citations, although Facebook mentions and tweets were also strongly correlated. In addition, original tweets and retweets were highly correlated with each other and with total tweets, suggesting that a single measure (total tweets) is sufficient. Other metrics showed only weak correlations with each other.

Dividing the reference Phase III sample into two subsamples according to the publication date provided by Altmetric Explorer revealed important differences (S1 Fig). The more recent

half of the publications (2H, after May 21, 2017) had higher mean Twitter, Wikipedia, and other counts, but lower Dimensions citations, than the older half (1H).

Three-factor analysis was conducted on the full range of metrics selected for inclusion (S3 Table). Two-factor analysis was also carried out on a subset of metrics excluding those with low incidence (policy document, PubMed guideline, and patent mentions) (S4 Table). These analyses revealed consistent groupings, such as Mendeley readers with Dimensions citations, and news, blog, and Wikipedia mentions.

## Weighting

Based on the results of these analyses and considerations, a framework for grouping metrics was developed comprising three component scores: social impact (news, blog, Twitter, Facebook, and Wikipedia mentions), scholarly impact (Mendeley readers, Dimensions citations, and F1000Prime posts), and societal impact (mentions in policy documents, PubMed guidelines, and patents). An initial statistical estimate of weightings was calculated as the inverse proportion of counts of each altmetric in the reference Phase III sample relative to the total number of all altmetric counts.

The stakeholder panel was conducted as part of a 1-day workshop with 14 stakeholders (9 female, 5 male), all employees of Novartis and representing different functions within the company (4 Medical Affairs, 9 Scientific Communications and 1 Commercial). Most (n = 12) were European, with 2 US representatives. Stakeholders were asked to summarize in one or two words and phrases what they felt 'impact' meant to them. 'Change' was a key theme, mentioned 5 times (twice in the context of changing clinical practice, once each for changing mindsets and changing dogma, as once as simply 'change'). 'Patients' were mentioned twice (in the context improving patient outcomes), while other phrases mentioned were communication, behaviour, access, reach, educate, utility, and supporting treatment decisions. Further discussions during the stakeholder panel meeting revealed the central importance given to guideline and policy document citations as a measure of article impact. This was also reflected in the quantitative session, in which guidelines and policy documents were allocated over one-third of the total points (Table 1).

Weightings derived from the statistical approach were revised to reflect findings from stakeholder value assessments. The selected weightings and their contribution to the total impact score based on the sample are shown in Table 2. In general, the approach taken was to balance the weighting such that the percentage contribution to scores in publications in the reference Phase III sample resembled the stakeholder value, while acknowledging relative importance

**Table 1. Value accorded to metrics by the stakeholder panel (quantitative scoring).**

| Metric | Social | Scholarly | Societal | Total | Percentage of points |
|---|---|---|---|---|---|
| **Twitter mentions** | 24 | 1 | 0 | 24 | 10 |
| **Facebook mentions** | 15 | 0 | 0 | 15 | 6 |
| **Blog mentions** | 16 | 0 | 0 | 16 | 7 |
| **News mentions** | 20 | 1 | 3 | 24 | 10 |
| **Wikipedia mentions** | 8 | 0 | 0 | 8 | 3 |
| **Dimensions citations** | 0 | 40 | 0 | 40 | 16 |
| **Mendeley readers** | 0 | 16 | 0 | 16 | 7 |
| **F1000Prime mentions** | 0 | 14 | 0 | 14 | 6 |
| **Guideline mentions** | 2 | 7 | 45 | 54 | 22 |
| **Policy mentions** | 0 | 0 | 31 | 31 | 13 |
| **Patent mentions** | 0 | 0 | 5 | 5 | 2 |

**Table 2. Weighting assigned to metrics included in the social, scholarly, and societal impact scores, along with their contribution to total impact scores in the reference sample.**

| Metric | Total in reference sample | Percentage of all metrics in reference sample | Social weighting | Scholarly weighting | Societal weighting | Percentage contribution to total in reference sample |
|---|---|---|---|---|---|---|
| **Twitter mentions** | 94,235 | 29.25 | 3 | – | – | 17.0 |
| **Facebook mentions** | 3821 | 1.19 | 3 | – | – | 0.7 |
| **Blog mentions** | 1086 | 0.34 | 10 | – | – | 0.7 |
| **News mentions** | 18,539 | 5.75 | 15 | – | – | 16.7 |
| **Wikipedia mentions** | 70 | 0.02 | 5 | – | – | 0.0 |
| **Dimensions citations** | 78,785 | 24.45 | – | 4 | – | 18.9 |
| **Mendeley readers** | 124,866 | 38.75 | – | 1 | – | 7.5 |
| **F1000Prime mentions** | 252 | 0.08 | – | 15 | – | 0.2 |
| **Guideline mentions** | 183 | 0.06 | – | – | 1800 | 19.8 |
| **Policy mentions** | 321 | 0.10 | – | – | 900 | 17.4 |
| **Patent mentions** | 59 | 0.02 | – | – | 300 | 1.1 |
| **Percentage contribution to total** | – | – | 35.1 | 26.7 | 38.2 | 100 |

(e.g. of news articles vs blogs) and prevalence (e.g. when Wikipedia entries were too infrequent to make a meaningful contribution without greatly inflated weighting relative to the value accorded by the stakeholder panel). To combine statistical and value-based weighting effectively, some related metrics were considered as combined entities (i.e. Twitter and Facebook mentions were allocated a combined 20% of points by stakeholders, and contributed a combined 17.7% to the total impact score in the reference Phase III sample).

## Predictor scores

The variance in total impact scores explained by each predictor score was moderately high (early predictor vs total impact score, $r^2 = 0.56$; intermediate predictor vs total impact score, $r^2 = 0.65$, S2 Fig). An overall predictor score can be calculated as the average of early and intermediate predictor scores. The variance in total impact scores explained by the overall predictor score was also moderate (overall predictor vs total impact score, $r^2 = 0.69$). Weightings calculated for each of the variables in the predictor score are shown in Table 3.

## Benchmarking

In total, 74 Phase III publications from the NEJM published in 2016 were identified for the benchmark sample. The non-adjusted, non-adjusted overall predictor score was selected as the

**Table 3. Weightings assigned to metrics included in the early and intermediate predictor scores.**

| Metric | Total in reference sample | Early predictor score | Intermediate predictor score | Percentage contribution to overall predictor score in reference sample |
|---|---|---|---|---|
| CiteScore | 15,384 | 57 | – | 26.2 |
| News mentions | 18,539 | 22 | – | 12.2 |
| Twitter mentions | 94,235 | 3 | – | 8.2 |
| Facebook mentions | 3821 | 30 | – | 3.5 |
| Mendeley readers | 124,866 | – | 12 | 44.8 |
| Blog mentions | 1086 | – | 125 | 4.1 |
| F1000Prime mentions | 252 | – | 146 | 1.1 |

Table 4. **Scores in the benchmark sample before and after benchmark adjustment.** Non-adjusted scores chosen as benchmarks are shown in bold.

| | Early predictor score | Intermediate predictor score | Overall predictor score[a] | Social | Scholarly | Societal | Total[b] |
|---|---|---|---|---|---|---|---|
| **Mean non- adjusted score** | 3218 | 4414 | 3816 | 1622 | 1593 | 2854 | 6068 |
| **Benchmark value** | 3816 | 3816 | 3816 | 2023 | 2023 | 2023 | 6068 |
| **Mean benchmarked score** | 84 | 116 | 100 | 80 | 79 | 141 | 100 |

[a]The overall predictor score is the average of early and intermediate predictor scores.

[b]The non-adjusted total impact score is the sum of the social, scholarly, and societal impact scores. The adjusted total impact score is the average of the adjusted component scores.

benchmark for predictor scores, and the non-adjusted total impact score was selected for total, social, scholarly, and societal impact scores (Table 4).

Dividing the non-adjusted total benchmark by 3 before applying it to the component scores had the effect of upscaling them so that the adjusted total impact score represents the mean of the components (rather than the sum, as in the non-adjusted total impact score). EMPIRE Index scores are calculated by dividing the unadjusted score of interest by the appropriate benchmark and multiplying by 100.

## Final scoring framework

An overview of the final EMPIRE Index framework is shown in Fig 2. The framework comprises the three component scores (social, scholarly, and societal impact), which are averaged to provide a total impact score. Each component score incorporates a separate type of altmetric, indicating a different aspect of engagement with the publication. The framework also includes the two predictor scores.

## Characterization of the EMPIRE Index

**Characterization in samples used in development.** The distributions of scores in the reference sample 1H and 2H, and in the benchmark sample, are shown in Fig 3. Of note, social impact scores were lower and societal impact scores were higher in 1H than in 2H. Predictor scores were higher than total impact scores in the reference Phase III sample but not in the benchmark NEJM Phase III sample, and median social impact scores were closer to median

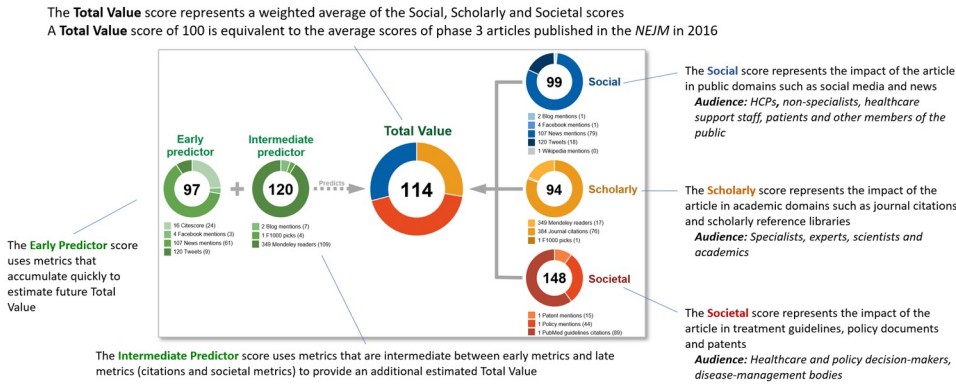

**Fig 2. Example of the EMPIRE Index score for a single publication.** HCP, healthcare provider; NEJM, *New England Journal of Medicine*.

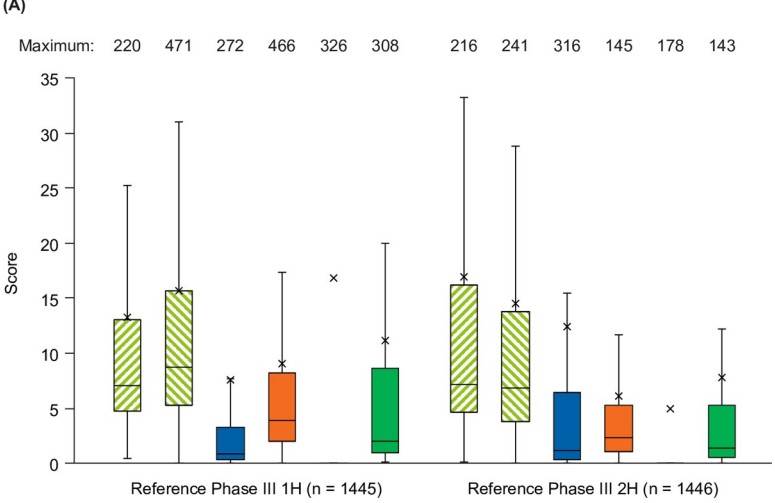

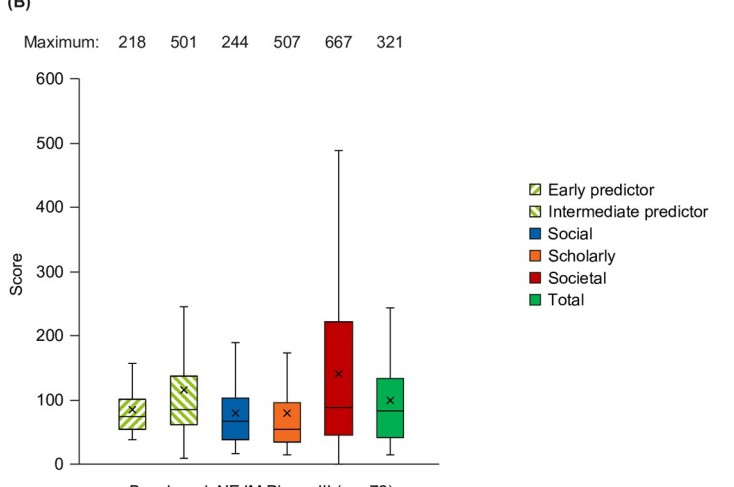

**Fig 3. Distribution of scores.** Distribution of scores in (A) the reference Phase III sample and (B) the benchmark NEJM Phase III sample. Box = 1Q–2Q, whiskers = 1.5 × interquartile range, X = mean. 1H, older 50%; 2H, younger 50%. NEJM, *New England Journal of Medicine*.

total impact scores in the benchmark NEJM Phase III sample than in the reference Phase III sample.

The correlations between component scores, the AAS, and CiteScore are shown in Table 5. Correlations between component scores were relatively low, the greatest being between social and scholarly impact scores. The social impact score correlated strongly with AAS, and both social and scholarly impact scores correlated modestly with CiteScore. However, the societal impact score is quite distinct from AAS, CiteScore, and the other component scores. Although predictor scores were moderately successful at predicting the total impact score, they were only weakly related to the societal impact score.

## Characterization in prospectively collected samples

**1-year update Phase III sample characterization.** Publication dates obtained from Alt-metric Explorer indicated that 194 articles were published prior to May 1, 2015; 1173 from

**Table 5. Correlations (Spearman r) between component scores, AAS, and CiteScore in the reference Phase III sample.** Correlations > 0.6 are shown in bold.

| Score | Early predictor | Intermediate predictor | Social | Scholarly | Societal | Total | AAS | CiteScore |
|---|---|---|---|---|---|---|---|---|
| Early predictor | – | 0.61 | 0.79 | 0.63 | 0.19 | 0.70 | 0.76 | 0.91 |
| Intermediate predictor | 0.61 | – | 0.59 | 0.87 | 0.26 | 0.76 | 0.59 | 0.55 |
| Social | 0.79 | 0.59 | – | 0.59 | 0.18 | 0.74 | 0.95 | 0.58 |
| Scholarly | 0.63 | 0.87 | 0.59 | – | 0.32 | 0.84 | 0.59 | 0.58 |
| Societal | 0.19 | 0.26 | 0.18 | 0.32 | – | 0.55 | 0.27 | 0.16 |
| Total | 0.70 | 0.76 | 0.74 | 0.84 | 0.55 | – | 0.78 | 0.57 |
| AAS | 0.76 | 0.59 | 0.95 | 0.59 | 0.27 | 0.78 | – | 0.56 |
| CiteScore | 0.91 | 0.55 | 0.58 | 0.58 | 0.16 | 0.57 | 0.56 | – |

AAS, Altmetric Attention Score.

May 1, 2016 to April 30, 2017; 1101 from May 1, 2017 to April 30, 2018; and 423 from May 1, 2018 to April 30, 2019. The drop in publication numbers in the latter period most likely reflects a lag in MEDLINE indexing. The 2019–2020 search identified 503 publications, of which 435 met the date criteria based on publication dates obtained from Altmetric Explorer. Mean EMPIRE Index scores in these year groups in both the original altmetric acquisition and the 1-year update are shown in Fig 4. Little change was found in the social impact component. Scholarly impact and, especially, societal impact continued to accumulate. The greatest increase in scholarly impact was seen in the most recent publications, while societal impact scores increased similarly across all 3 years sampled.

**NEJM notable articles characterization.** In total, 48 notable articles were identified by NEJM editors from 2016 to 2019. Mean impact scores from the 2016 subset are shown in Fig 5, with mean scores from the 2016 benchmark NEJM Phase III sample for comparison. Notable articles had higher social and societal impact than benchmark articles.

Of the 48 articles, the focus was assessed to be interventional in 24 cases, observational in 10 cases, innovative in 6 cases, and surgical in 8 cases. After adjusting for publication year, observational studies were found to have the highest total impact, with other publication types having lower impact scores across all component scores except for the societal impact of surgical studies. Innovative studies had notably low societal impact, indicating that they were infrequently referenced in guidelines or policy documents (Fig 6).

**Case examples.** We present here two illustrative case examples. The first was the publication of VERIFY, a Phase 3 study of vildagliptin in patients with Type 2 diabetes, published in The Lancet on the 18th September 2020 [41]. The initial analysis, conducted on the 7th April 2020, was 202 days after publication, by which time it had gained notably high Early Predictor Score and so was selected for further investigation (Table 6). The publication had been timed to coincide with presentation at Annual Meeting of the European Association for the Study of Diabetes (EASD) in Barcelona, Spain (16–20 Sept, 2019) and was accompanied by press releases from the EASD and Novartis. The Lancet also tweeted, but this was accompanied by a limited number of retweets. The article had been picked up in guidelines at an early stage, and subsequent tracking identified increases in the societal impact score due to guidelines citations and a patent citation. The scholarly impact score was lower than predicted at the time of initial assessment. On the most recent follow up (3 September 2021) the scholarly impact score had increased to 23.

The second case example is the publication of Two Phase 3 Trials of Inclisiran in Patients with Elevated LDL Cholesterol, published in the NEJM on 18 March 2020 [42]. At the time of

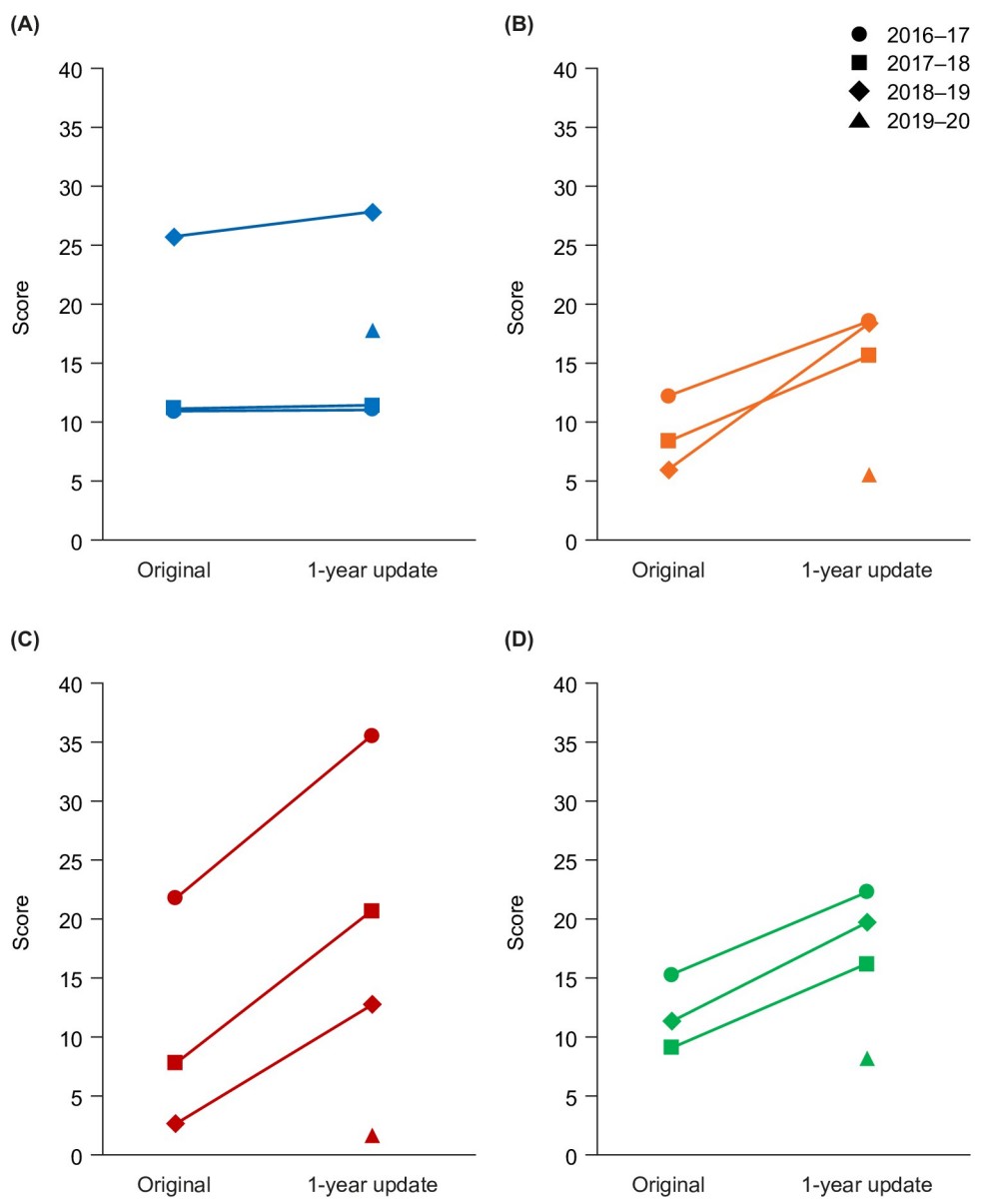

**Fig 4. Mean impact scores in the original reference Phase III sample and the 1-year update Phase III sample.** (A) Social, (B) scholarly, (C) societal, and (D) total mean impact scores.

analysis, only 20 days after publication, it had achieved a very high Early Predictor Score, due to a large number of news articles linked to a Novartis press release. Follow up analyses showed that the Social score continued to increase as a result of an NEJM infographic that was tweeted by an academic expert with 8,000 followers, and a postdoctoral student using the paper as an example of best practice in data visualization. On the most recent follow up (3 September 2021) the scholarly impact score had increased to 50, commensurate with predictions suggesting that impact among the academic community had been satisfactory. However, it had not yet achieved detectable societal impact, suggesting that impact on clinical practice may be limited.

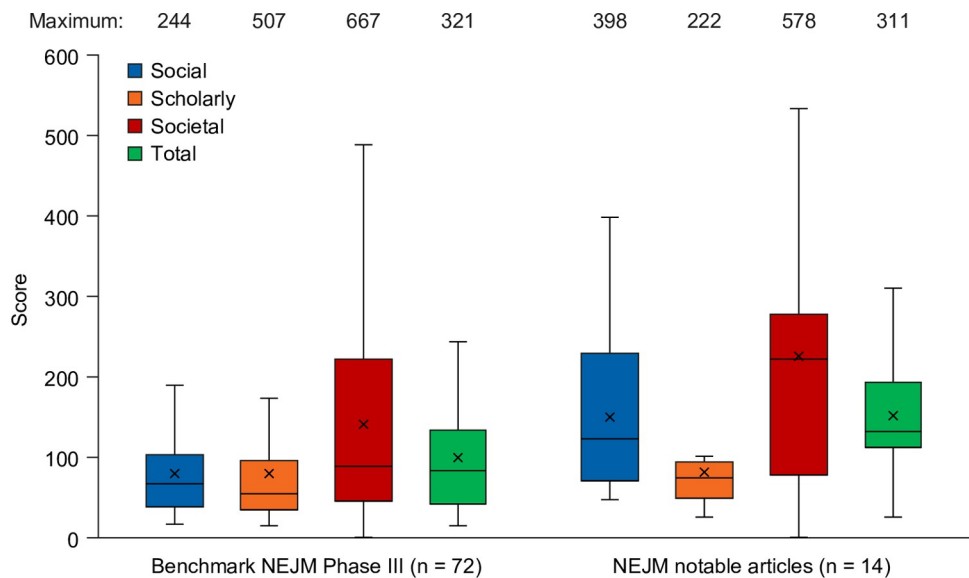

**Fig 5. Mean impact scores for NEJM notable articles from 2016 compared with benchmark scores.** Box = 1Q–2Q; whiskers = 1.5 × interquartile range; X = mean. NEJM, *New England Journal of Medicine*.

## Discussion

We have developed the EMPIRE Index, a metric framework to assess the multidimensional impact of medical publications, including the potential impact on clinical practice. It avoids the pitfalls of JIF-based research assessment and unidimensional scoring systems. It also fulfills the Leiden criteria of being open, transparent, and simple [43].

The EMPIRE Index aggregates selected article-level metrics into meaningful component scores and weights them according to the value placed on them by members of a stakeholder

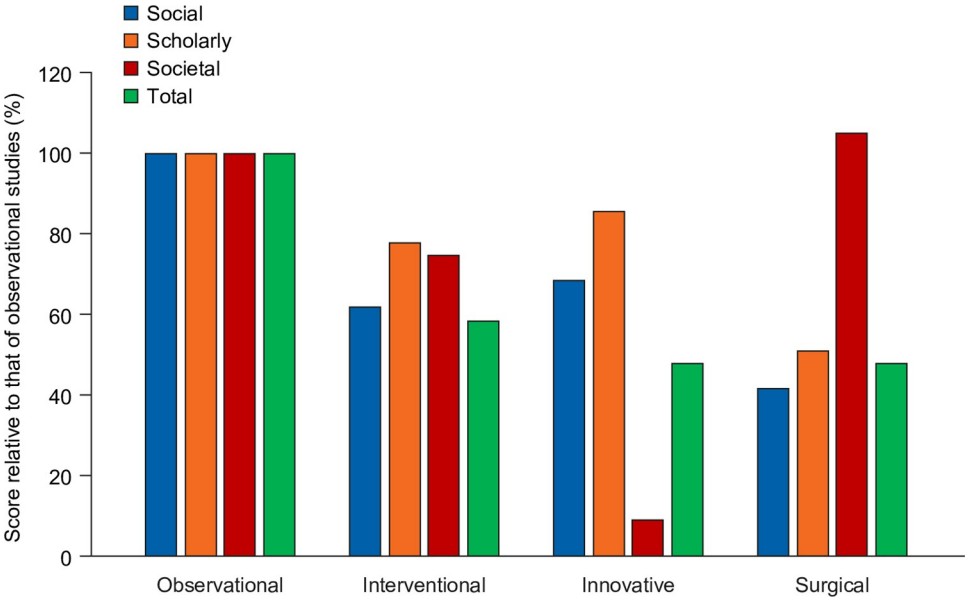

**Fig 6. EMPIRE Index scores for NEJM Notable articles.** EMPIRE Index scores are expressed as percentages of the scores achieved by observational studies.

**Table 6. EMPIRE Index scores for the two case example publications and time of initial analysis and subsequent follow-up.**

| Publication | Publication age at time of analysis (days) | Early reach | Intermediate reach | Social | Scholarly | Societal | Total |
|---|---|---|---|---|---|---|---|
| VERIFY | 202 | 37 | 23 | 34 | 5 | 89 | 43 |
| VERIFY (follow up) | 415 | 39 | 41 | 36 | 12 | 267 | 105 |
| "Two Phase 3 trials" | 20 | 50 | 0 | 36 | 0 | 0 | 12 |
| "Two Phase 3 trials" (follow up) | 233 | 60 | 58 | 51 | 19 | 0 | 23 |

panel and statistical analysis of a representative sample of articles. It differs conceptually from both the AAS and other, recently developed scoring systems: the #SoME_Score [44], the Weighted Altmetric Impact, and the Inverse Altmetric Impact [18, 38]. First, the value-based approach to the weighting and grouping of metrics recognizes that simple statistical associations may be sample-dependent and may not relate to underlying conceptual underpinnings. Second, the EMPIRE Index is specifically designed for medical publications. Many studies have documented different scales and relationships between metrics in various disciplines [15, 20, 24, 25] and, given that the value of each metric is inherently subjective, this value is unlikely to be consistent across scholarly disciplines. Third, the EMPIRE Index is scaled against a clearly defined, relevant benchmark, because interpretation of a novel composite metric is difficult without such a reference point.

Such are the potential advantages of the EMPIRE Index. However, its utility is dependent on the robustness of the selection grouping, weighting of metrics, and benchmarking, as well as its performance in the evaluation of suitable publications. In the process of investigating these factors, a series of results of broad interest to the altmetrics community were generated. These will be discussed in the sections that follow.

## Metric selection

Suitable metrics were identified for inclusion by reviewing the coverage and density of metrics obtained for the reference Phase III sample through the two most established metric providers: Altmetric and PlumX. Previous research has shown significant differences between these providers in terms of Mendeley readers and Twitter coverage, as a result of different approaches to collecting, tracking, and updating metrics [45–47]. Furthermore, the approach to covering news and blog posts differs greatly between PlumX and Altmetric Explorer [48]. We found broadly similar metrics between the two providers except for news articles, with Altmetric reporting twice as many for our sample as PlumX, and Facebook mentions (because Altmetric extracts only mentions on Facebook pages whereas PlumX also extracts 'likes').

The reference dataset was selected to provide a sample rich in altmetrics. PlumX identified at least one metric for 99% of our sample, while the figure was 83% for Altmetric. This result compares favorably with that of previous work [17, 20, 24, 25, 49, 50], most likely indicating the increasing volume of altmetric activity. One important metric not included was article views and downloads. Although these data were provided by PlumX, we found them to be patchy, with many articles reporting metrics such as tweets or Mendeley readers but no page views or downloads on the EBSCO information service.

Similar to previous investigators, we found that news, blog, Twitter, and Facebook mentions, Mendeley readers, and Dimensions citations were the most common metrics in our sample. These metrics were included in our analysis, as well as additional metrics that, although rare, provided valuable insights into article impact: citations in policy documents, guidelines, patents, Wikipedia, and F1000Prime.

## Rationale for the three component scores

The EMPIRE Index comprises three component scores, each representing a different factor underlying the observed patterns of metrics seen in the reference Phase III sample. The social impact component represents actions that involve or are accessible to the general public as well as healthcare professionals and academics. The scholarly impact component represents actions with an academic focus. The societal impact component represents actions in which the publication has been used to inform decisions around optimal care or, in the case of patents, medical advances.

To inform our metric grouping, correlation analyses were performed and exploratory factor analysis was used. These revealed a close connection between Mendeley readers and Dimensions citations, in line with findings from previous research [51–53]. Correlations were also found between Twitter and Facebook mentions, and news/blog and social media mentions, which again aligns with previous observations [52, 53].

No meaningful correlations were found between mentions in F1000Prime articles, policy documents, guidelines, patents, or Wikipedia articles and other metrics. These metrics have not previously been widely studied, and the low correlations observed may reflect their very small coverage–over 90% of publications score zero on these metrics. However, an analysis of four biomedical journals found that recommended articles are cited more frequently than non-recommended articles from the same journal [54], while Bornmann and Haunschild (2018) reported that F1000Prime recommendations were more closely correlated with Mendeley readers and Dimensions citations than with Twitter mentions [55].

Pairwise correlations can give useful insights into relationships between different metrics, but for the purposes of reducing data into composite scores it is helpful to understand the shared variance between multiple metrics. The exploratory factor analysis in our study produced findings consistent with those reported in previous literature [15–18]. Separating articles into those that were older (1H) and younger (2H) showed that citations (including policy and guideline mentions) and Mendeley readers consistently grouped into one factor; news, blogs, Wikipedia, and F1000Prime mentions grouped into a second factor; and Twitter (and, usually, Facebook) mentions comprised a third factor. A two-factor analysis excluding policy document, guideline, and patent mentions confirmed that Mendeley readers and Dimensions citations formed a separate group from the remaining metrics.

Separating metrics into statistically and conceptually homogenous groupings meets a key criterion specified by Gingras and Larivière for well-constructed indicators [56]. Another criterion they specify is that an indicator should adequately represent the concept that it is intended to represent. Each altmetric represents a different action on the part of an audience; this has implications for how we understand the meaning of individual metrics [7] and whether these statistical associations represent meaningful groupings.

The social impact component comprises tweets, Facebook likes, blog and news article mentions, and Wikipedia citations. Much remains unknown about the motivation for tweeting, given that most tweets are empty of context [57] and content [58]. Often all that is certain is that the tweeter felt the research interesting enough to broadcast. Social media platforms are known to be used mostly by the general public, so a central motivation for scholars to tweet is likely to be to communicate and explain their work to lay people [59]. This may be particularly true of publications in biomedical sciences, which attain greater Twitter interest than those in other scholarly disciplines [59]. Twitter communities linked through publication tweets tend to be led by organizational accounts associated with well-known journals or leading scholars [60], although at least half of sharing on social media is likely to be non-academic [61, 62]. An analysis of Facebook users who shared links to medical and health-related research articles

found that more than half were not academic, while 16% were healthcare professionals and only 4% were academics [62]. Similarly, blogs and news articles are likely to be read by a mix of audiences, including (for biomedical articles), healthcare practitioners and patients. News article metrics are derived from a curated list of news sources, including general interest, local interest, science/technology and health sciences outlets [48]. It should be noted, however, that the curated list for Altmetric.com data are biased towards the English language and around 50% are located in the USA [48]. Wikipedia is a widely used source of scientific information, including by scientists [63], and it is one of the most widely read accessed sources of medical information by the general public [64]. Thus, social media activity, news articles and Wikipedia share some commonality in that they play a role in disseminating information across diverse audiences.

The Scholarly Impact component comprises scholarly citation, reference manager data, and F1000Prime recommendations. Citations indicate that one scholarly work has been acknowledged by another. Conventionally this is seen as an indication of influence or impact, although the act of citing is complex and can be influenced by a range of factors such as post-hoc justification for a research project [65]. Nevertheless, an analysis of 640 highly-cited medical publications found that only 9% were also found in a list of 652 articles with the highest AAS (i.e. primarily social media and news mentions), suggesting distinct motivations for scholarly citing versus sharing across diverse audiences [22]. Mendeley saves require the reader to have access to the free-to use Mendeley reference manager platform, and so reflect useage among individuals who consume a lot of scholarly literature. Although Mendeley users often add articles to their library with the intention of citing them, many also add these for professional or teaching purposes, which may explain why some articles have many readers but few citations [55]. Mendeley saves therefore have been suggested as an alternative to download counts as a source of readership evidence [6], although limited to those readers who have a Mendeley account. F1000Prime recommendations indicate an article has been recommended by F1000Prime Faculty members who have been nominated by peers as experts in their fields. Interestingly, articles rated in F1000Prime reviews as a 'technical advance' received higher Mendeley scores, but not higher Twitter scores, than those that were not rated this way [66]. The reverse was true for articles considered a 'good for teaching', further underscoring the difference between Mendeley and Twitter indicators.

The societal impact score comprises citations in medical guidelines published in peer-reviewed journals and indexed on PubMed, policy documents (i.e. grey literature, typically in the medical arena these will be technical assessments of products as part of guidelines development) and patents. These represent a different activity from citations in scholarly literature, since only guidelines/policy documents and patent citations clearly reflect wider societal impact [67, 68]. This is supported by our finding that NEJM notable articles score higher in societal impact relative to scholarly impact compared with typical Phase III clinical studies publications. It should be noted, however, that guidelines do not always contain references (although these may be provided in associated grey literature) and, when present, these references do not explicitly indicate their value to the guideline [69].

## Weighting

The weighting of metrics in the EMPIRE Index was based on three considerations: the prevalence of metrics in the reference sample (highly prevalent metrics were weighted less), the need for each component to make a substantial contribution to the total impact score, and the value given to each metric as an indicator of impact. As a result, the weighting is quite different from other approaches based on purely statistical considerations.

Several approaches have determined weighting by regressing altmetrics on citations. These typically result in, for example, higher weighting given to blog posts and Mendeley readers than to news articles (because blog posts are relatively uncommon) [20, 44, 52, 70]. Because the target variable is journal citations, each Mendeley save or F1000 citation may be weighted in a similar way to or higher than a policy document citation [44, 71]. Ortega has developed weightings based on principal component analysis and also on inverse prevalence (so that the rarest metrics receive the highest weighting). The two approaches create very different weightings–for example, a news article carries half the weight of a publication citation in the Weighted Altmetric Impact, but eight times the weight of a publication citation in the Inverse Altmetric Impact [23, 72]. These statistical approaches give very different results from the weighting developed for the EMPIRE Index.

## Predictor scores

Given that some altmetrics accumulate early, there is long-standing interest in the use of a limited selection of rapidly accumulating altmetrics to identify publications likely to have high long-term impact. Earlier work has employed multivariate regression with citations as a measure of long-term impact [12, 20, 44, 70, 71, 73] but, as we have seen, citations are only one of several measures of long-term impact.

Among common metrics, tweets and news articles accumulate most rapidly after publication, while Mendeley readers, blogs, and F1000Prime articles increase more gradually [6, 39, 40]. Wikipedia and policy document mentions can, like article citations, take well over a year to accumulate [39, 74]. The EMPIRE Index addresses this by using two predictor scores–early and intermediate.

The early predictor score also uses CiteScore, a journal-level metric similar to Journal Impact Factor. Because CiteScore is not an article-level metric it is not suitable for assessing the impact of individual articles. However, the choice of journal can have a significant effect on the impact of the publication, primarily because of readership (i.e. some journals have significantly higher reach into key audiences). Unfortunately, there is no consistent, publicly available measure of journal reach measure, because most publishers don't make readership figures available in a comparable format. We therefore included Citescore not as a measure of impact, but as a proxy for journal reach and therefore as a partial predictor of likely future impact.

CiteScore, in this context, can be thought of as a proxy for the exposure an article is likely to have; it has previously been shown that combining citations over the first year with JIFs accurately predicts future citations [74, 75].

Predictor scores are a purely statistical construct so the weighting is quite different from the EMPIRE Index itself; however, the weighting is also different from methods employed in previous work using citations as a target. Compared with studies mentioned earlier that used statistically based weighting with only citations as a target, in the EMPIRE Index predictor scores, Mendeley readers carry less weight relative to news article citations. This most likely reflects the broader basis of the EMPIRE Index compared with citation-only targets.

The reasonably strong relationship between predictor scores and the total impact score in the reference Phase III sample is to be expected, given that they share many of the same metrics. However, the weak correlation with the societal impact score indicates that the predictor scores will lack precision in identifying high-impact publications (given the importance of the contribution of societal impact to the total impact). Further work using longitudinal datasets is required to improve these predictor scores.

## Responsiveness and characterization

The responsiveness and utility of the EMPIRE Index was evaluated in several ways. Averages and distributions of scores in the reference Phase III sample and the benchmark NEJM sample were explored, showing that both samples had similar social and scholarly metrics and the latter had far higher societal metrics. Because the scores were scaled to the benchmark NEJM sample, this resulted in predictor scores lacking sensitivity for lower-impact publications (i.e. although they retained precision for identifying higher-impact articles, they tended to overpredict the impact of lower-impact articles uniformly).

The social score was shown to be closely correlated with the AAS. The AAS weights metrics in a way that is not possible for users of the Altmetric Explorer dashboard–news outlets are weighted in a proprietary (and undisclosed) tier system, while retweets are assigned only 75% of the weight of original tweets [9]. The high correlation between the social score and the AAS thus reassures users that these nuances make little difference.

Changes over time were evaluated in a 1-year follow-up of the reference Phase III sample. The minimal change in the social impact component further underlines the similarity of this component to the AAS, and supports the notion that news article and tweet metrics accrue soon after publication. Both scholarly and societal impact scores continued to increase, and further follow-up is needed to identify the point at which these scores plateau.

Finally, an independent dataset was investigated: articles selected by NEJM editors for their practice-changing potential. These papers had substantially higher societal impact than the benchmark set of NEJM Phase III articles, supporting the sensitivity of the societal impact component in identifying practice-changing publications. Furthermore, innovative articles were found to have relatively low societal impact, indicating that although these are of interest to scholars and wider society, they do not directly feed into clinical practice changes. Conversely, articles on surgery had a high impact on practice even though social and academic interest was low.

## Using the EMPIRE Index

The EMPIRE Index can be used to monitor large numbers of publications (for example, relating to a research project or clinical trial programme) to assess whether the publications are having the hoped-for impact. It can be used to identify publications that are having higher than expected impact, with implications for best practice in publication dissemination (for example, whether enhanced publication activities such as videos or infographics affect publication metrics). It could also be used to identify publications with lower-than-expected impact, which could signify that additional communication efforts are needed to reach audiences that may be interested in the topic (or that the topic is of low interest to the audiences concerned). The EMPIRE Index can also be used on large datasets, for example to see how different journals are associated with different kinds of impact, which could inform journal choice for submission.

## Weaknesses

Although the EMPIRE Index provides advantages over existing metric approaches, it has some potential weaknesses. For example, grouping and value weighting have a large subjective component that may not reflect the value assigned to metrics by others. However, the transparent nature of the approach will hopefully stimulate further debate and discussion around the inherent subjectivity and allow for future refinements.

The analyses conducted were based on a closely defined subset of medical publications, in terms of both content (Phase III trials) and publication date. As metrics evolve over time

owing to changes in the way audiences engage with publications or technical advances in the way metrics are recorded, these original analyses and assumptions may not apply. They may also not apply to other publication types or study designs and may vary across disease areas. Predictor scores are based on results of cross-sectional, rather than longitudinal, analyses; further follow-up will allow these scores to be refined and improved. Furthermore, benchmarking to very high-impact articles results in predictor scores that tend to overestimate the final impact of more usual articles.

Any indicator must represent a relatively homogenous construct to be considered meaningful [56]. The component scores of the EMPIRE Index (Social, Scholarly, and Societal Impact) have been specifically designed to meet this criterion, but the combined Total Impact score inevitably does not. Interpretation of the Total score is difficult if quoted in the absence of supporting component scores.

Lastly, although the scoring system is transparent and reproducible, it depends on metrics aggregated by two different proprietary systems. These metrics may not be available to all intended users of the index.

## Conclusions

The EMPIRE Index is a novel metric framework incorporating three component scores that respond to different types of publication impact: social, scholarly, and societal. Whereas the social impact score is similar to the AAS and the scholarly impact score is closely linked to (but broader than) article citations, the societal impact score reflects a key and distinct aspect of publication impact. In a similar way to the AAS, the EMPIRE Index weights metrics subjectively to reflect their value from the user's perspective as well as by prevalence. Unlike the AAS, it is designed for a limited subject area (medicine) and weights and benchmarks the metrics accordingly. It also has a clear, transparent explanation of the scoring system, and provides predictor scores to give an early estimate of likely future impact.

The development of the EMPIRE Index incorporates objective analysis and subjective values. It is, therefore, only directly relevant to stakeholders who share broadly similar perspectives to our panel. However, the process used for developing the EMPIRE Index is general utility; any interested party can reweight the EMPIRE index using subjective values arrived at using their preferred method.1

Several potential uses are envisaged for the EMPIRE Index. Because it provides a richer assessment of publication value than standalone traditional and alternative metrics, it will enable individuals involved in medical research to assess the impact of related publications easily and to understand what characterizes impactful research. It can also be used to assess the effectiveness of communications around publications and publication enhancements such as infographics and explanatory videos. Fuller validation of the EMPIRE Index requires additional prospective and cross-sectional studies, which are ongoing.

## Supporting information

**S1 Table. Summary statistics for metrics obtained via (A) Altmetric, (B) PlumX, and (C) journal-level, citation-based indices.** Coverage is the proportion of articles with > 0 on that metric.
(DOCX)

**S2 Table. Correlations (Spearman's r) between investigational metrics in the sample of Phase III clinical trial publications.** Correlations > 0.5 are shown in bold.
(DOCX)

**S3 Table. Three-factor analysis of included metrics in (A) the full sample, (B) older papers (1H), and (C) younger papers (2H).** Highest loadings for each metric are shown in bold.
(DOCX)

**S4 Table. Two-factor analysis of metrics excluding citations in policy documents, PubMed guidelines, and patents in (A) the full sample, (B) older papers (1H), and (C) younger papers (2H).** Highest loadings for each metric are shown in bold.
(DOCX)

**S1 Fig. Percentage change in mean publication metrics in the more recent half of the publications (2H) versus the older half (1H).**
(TIF)

**S2 Fig. Correlation of total impact scores with (A) early predictor and (B) intermediate predictor scores.** Scores shown are not adjusted to the benchmark.
(TIF)

## Acknowledgments

The authors thank Heather Lang of Oxford PharmaGenesis, Oxford, UK for providing valuable insights in the initial design phases of the project and for co-facilitating the stakeholder insights workshop. Editorial support (manuscript proofreading, figure drawing, and project management) was provided by Oxford PharmaGenesis.

## Author Contributions

**Conceptualization:** Avishek Pal.

**Data curation:** Tomas James Rees.

**Formal analysis:** Tomas James Rees.

**Funding acquisition:** Avishek Pal.

**Investigation:** Avishek Pal, Tomas James Rees.

**Methodology:** Avishek Pal, Tomas James Rees.

**Writing – original draft:** Tomas James Rees.

**Writing – review & editing:** Avishek Pal.

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
