## [Decision Letter · Decision Letter 0]

27 Aug 2021

PONE-D-21-11452

Introducing the EMPIRE Index: A novel, value-based metric framework to measure the impact of medical publications

PLOS ONE

Dear Dr. Rees,

Thank you for submitting your manuscript to PLOS ONE. After careful consideration, we feel that it has merit but does not fully meet PLOS ONE’s publication criteria as it currently stands. Therefore, we invite you to submit a revised version of the manuscript that addresses the points raised during the review process.

The reviewers have raised a range of questions and concerns regarding the presentation of your methods and interpretation of your data that need to be carefully addressed when preparing your revisions.

We look forward to receiving your revised manuscript.

Kind regards,

Jamie Males

Staff Editor

PLOS ONE

1. Please ensure that your manuscript meets PLOS ONE's style requirements, including those for file naming. The PLOS ONE style templates can be found at https://journals.plos.org/plosone/s/file?id=wjVg/PLOSOne_formatting_sample_main_body.pdf and https://journals.plos.org/plosone/s/file?id=ba62/PLOSOne_formatting_sample_title_authors_affiliations.pdf.

“I have read the journal's policy and the authors of this manuscript have the following competing interests: Avishek Pal is an employee of Novartis Pharma AG. Tomas Rees is an employee of Oxford PharmaGenesis and received research funding from Novartis Pharma AG for this study.”

Additional Editor Comments (if provided):

Reviewers' comments:

Reviewer's Responses to Questions

**Comments to the Author**

1. Is the manuscript technically sound, and do the data support the conclusions?

Reviewer #1: Yes

Reviewer #2: Yes

Reviewer #3: Partly

2. Has the statistical analysis been performed appropriately and rigorously? 

Reviewer #1: Yes

Reviewer #2: Yes

Reviewer #3: Yes

3. Have the authors made all data underlying the findings in their manuscript fully available?

Reviewer #1: Yes

Reviewer #2: Yes

Reviewer #3: Yes

4. Is the manuscript presented in an intelligible fashion and written in standard English?

Reviewer #1: Yes

Reviewer #2: Yes

Reviewer #3: Yes

5. Review Comments to the Author

Reviewer #1: Hi Dear Sir/Madam,

The article is appropriate for publication in this format. It just need to promote statement of the problem as well and also method must be repeatable for all researcher.

Sincerely Yours,

Reviewer #2: Interesting paper in which the authors propose a composite indicator to assess medial clinical trial publications' societal, social and scientific impact.

I have several concerns that should be addressed before considering the paper worth publishing.

MAJOR COMMENTS

Although the authors acknowledge the critiques to composite indicators (problems with interpretation, misleading views, etc.) they do not substantiate the need for the indicator they propose, nor the utility. Wouldn't readers be able to see what the EMPIRE Index offers by simply looking at the individual metrics included? I suggest the authors to read this paper where theoretical problems with the use of composite indicators are exposed (in this case with regard to the H-Index) https://doi.org/10.1002/asi.21678

I find troubling the uncritical way with which altmetric indicators are embraced, you indicate that Twitter, Facebook, etc. relate to social impact. What do you mean by social impact? In the same vein, you indicate that policy documents refer to societal impact. What is the difference between these two types of impact? Do mentions in these venues reflect impact? On whom? I think the paper would really benefit from a theoretical framing and motivation which would explain many of the decisions which are later made.

You discuss the value of metrics and how these are subjective. But wouldn't that invalidate your whole method? If you were to replicate the same exercise with the same or a slightly different panel of stakeholders, would the weighting still hold?

Overall, I find the paper to be methodologically robust as to the analyses the authors do, but the authors make a series of interpretations from the data for which they provide no explanations and ignore much of the literature critical with the potential use of altmetrics to measure quantitatively impact. I think that by including a more reflexive and critical review of the literature and a theoretical framework by which they can revisit and sustain many of the claims they make the paper will be much stronger.

I suggest some papers which may be useful:

- On value of research: https://link.springer.com/content/pdf/10.1007/s11024-011-9161-7.pdf

- Critiques to the AAAS: https://doi.org/10.1007/s11192-016-1991-5

- On criteria for evaluating indicators: https://www.researchgate.net/profile/Nirmala-Svsg/post/What-indicators-applied-for-evaluating-online-catalogs-at-universities/attachment/59d63e0779197b807799ab81/AS%3A422227658186752%401477678326575/download/Criteria-MIT6.pdf

Reviewer #3: Overall, I'm intrigued by the proposal of this new metric, and think it could merit publication. It's certainly novel, and the statistical analysis seems appropriate (though I am far from the most qualified judge on this matter). The metric is thoughtful, and in theory, could give useful information about an article that we don't currently have. That said, I have some serious concerns about how this metric was both constructed and presented.

My biggest concern with this article is that it’s using (mostly) altmetrics as indicators of impact, and even of specific types of impact (I count 4 – predictive scholarly impact, along with social, scholarly, and societal). The correlation between altmetrics as a whole and their value toward quantifying impact has never been clearly laid out, and while there is some evidence to justify some of how it is being used (such as a high degree of correlation between Mendeley downloads and citations for STEM fields, justifying Mendeley download as a proxy for scholarly impact), there’s a lot that is still up for debate (what tweets signify, for example – for detail on that below). Altmetrics are commonly discussed as a complement to existing bibliometrics, because they serve to bolster and support existing impact claims rather than introduce their own – it’s why the words “attention” and “engagement” get used by Altmetric, among other stakeholders, when describing the value of altmetrics. They’re also commonly discussed for their qualitative value – being able to demonstrate moments of impact (say, research being tweeted by a relevant non-profit or titan of the field) rather than measure and contextualize their quantity (beyond the very high-level contextualization of an AAS compared to similar articles).

Becker Model – I’d like to see the Becker Model (https://becker.wustl.edu/impact-assessment/files/becker_model-reference.pdf) mentioned, since it is the major impact framework for biomedicine that goes beyond scholarly impact / citation-based evaluation. There are some areas of overlap with EMPIRE, but it’s clear that it was not consulted or used in any significant way. Any reason why?

Concerns about societal impact – I suggest that the measures included for this metric are too low to quantify, and correlation with benchmark articles could be due to attention related to those articles making them more attractive to policymakers, guideline producers, etc. Not everything that can be measured should be quantified, and this grouping makes an excellent case for qualitative description rather than quantification, since a single citation can make a big impact on the score. I know from personal experience that the “policy” category of Altmetric is something of a weak area, with links to documents that have no right to be called policy papers, so I’m wary of its use, and the lack of correlation of the societal measure with ANY other measure gives me pause. I’m not as familiar with guidelines, though I suspect that metric is a bit more straightforward and appropriate for the biomedical field.

Transparency – I’d like to see more transparency regarding how the weighting was established – was there good agreement between the stakeholders, especially among different stakeholder groups? EMPIRE hinges on proper weighting. There is some good discussion around weighting, but this group needs more description – how many, how the weighting process went and how many iterations there were, demographics such as age and gender to capture an appropriately diverse set of opinions. This is one of the first times we’re seeing altmetrics directly used as an IMPACT indicator, so I’d like to hear a LOT more about this discussion.

Predictors - I don’t take much issue with the predictive metrics, though just to note, Figure 2 incorrectly implies, through the arrow pointing to the total value, that the two predictors form the “total value” score – I wonder if maybe the three composite scores should be on the left, and the predictors should be on the right, or some other way to show that the predictor scores are separate. Anyway, my one significant issue is in using CiteScore, while also claiming that EMPIRE moves away from the problematic JIF. CiteScore is virtually identical to JIF, just with a different set of indexed journals (Scopus rather than Web of Science), but all of the existing biases/power dynamics, time restrictions, lack of complexity, and other JIF-related complaints still exist. I think EMPIRE should be entirely distinct from citation-based evaluations. As is, its inclusion just serves to reinforce journal-level metrics/classification rather than provide a distinct metric from them – with CiteScore, it becomes less about the attention an individual article is receiving and more about the prestige, notoriety, and reach of a publication. I don’t like that the two are just casually mixed together, and think it’s a stronger metric, and more accurate gauge of early attention at the article level, without it.

I’d like to hear more about the intended use of this metric, specifically from Novartis Pharma AG and Oxford PharmaGenesis, as well as the intended audience for this metric. I greatly appreciate the weaknesses section of this paper, including mention of proprietary subscriptions necessary to calculate, but as with any new metric, the potential for misuse is awfully high for this. I think adding intended or suggested case uses and/or limitations would help with this – specific article types are mentioned, for example, but it should be clearly stated that this isn’t applicable outside of the biomedical field. Are EMPIRE scores meant to be directly compared to establish impactful articles along multiple impact dimensions? Are they used to justify the impact of a single researcher or research facility’s research? Further, it should also be stated more clearly that this metric is just an attempt at quantifying a much more complex concept, and should not be used as a sole metric for research evaluation. As mentioned above, some of the conjectures about what specific altmetrics like tweets signify are overly general and are not well agreed-upon (some tweets may be with an aim to influence a more general public, but academic Twitter is very much a thing, so you could also consider tweets as measures of scholarly or practitioner impact, depending on the context – in short, we’re still not sure what tweets mean, which is why altmetrics don’t usually get directly associated with impact metrics at all). Going back to the Becker Model, there’s SO much incorporated into its model that can’t be quantified, and some of the indicators in EMPIRE aren’t included there. In short, there’s a great deal of uncertainty that remains with EMPIRE, mainly due to assumptions about what individual altmetrics signify.

In short, many aspects of EMPIRE are highly subjective. If Novartis Pharma AG and Oxford PharmaGenesis want to use it for internal evaluation purposes, that’s their prerogative, but I have deep concerns about this, and feel like there are too many assumptions being made about its validity to warrant general use without a lot more explanation about its intended purpose and large limitations.

6. PLOS authors have the option to publish the peer review history of their article (what does this mean?). If published, this will include your full peer review and any attached files.

Reviewer #1: No

Reviewer #2: No

Reviewer #3: No

---

## [Author Response · Author response to Decision Letter 0]

15 Sep 2021

We have provided a point-by-point to the reviewer comments in our 'response to reviewers' letter. Our responses are reproduced below (I haven't included the reviewer comments that these are a response to, as it made it difficult to read). 

Reviewer #1

We thank Reviewer #1 for their comments. We believe that the manuscripts states the problem clearly and that the methods are repeatable for all researchers. 

Reviewer #2

We believe that the popularity of summary scores such as the Altmetric Attention Score (AAS) demonstrates the interest in and potential utility of summary metrics scores, and the critiques of the AAS mentioned in our paper and also by the reviewers illustrates the need for an improved approach. 

Demonstrating the utility of the approach will depend upon use cases. These are outside the scope of this initial methodological paper, although we do have follow-up investigations in preparation for publication. 

The EMPIRE Index is not a raw count of metrics but weights them, so provides information not immediately discernible from raw counts. Furthermore, we developed the EMPIRE Index because we find people are often overwhelmed by the number of individual metrics. That is why, for example, the AAS is so popular. EMPIRE Index is, in essence a data reduction tool to make the individual metrics more digestible, but still retaining enough nuance to make different types of impact visible. 

The h-index is conceptually different from the EMPIRE Index because it depends on the interaction between two metrics. That is, it is not a homogenous indicator, in the terminology introduced by Gingras and Larivière (in the reference provided by this reviewer: Beyond Bibliometrics: Harnessing Multidimensional Indicators of Scholarly Impact) This causes it to have all kinds of problematic ratio and scalar properties, as noted in the referenced article. 

The EMPIRE Index component scores are composed of related metrics aggregated linearly. This means that simple processes such as ranking, adding and averaging perform as expected, unlike the h-index. 

However, the EMPIRE Index is susceptible to other weaknesses, as noted in our paper and by the reviewers. In particular, as noted in reference 7 (Copiello S. Multi-criteria altmetric scores are likely to be redundant with respect to a subset of the underlying information. Scientometrics. 2020;124: 819–824. doi:10.1007/s11192-020-03491-9) and by by Gingras and Larivière. We developed the EMPIRE Index with component scores of closely related metrics to minimise this problem, however the total score is inevitably susceptible. We have expanded the ‘Weaknesses’ section to acknowledge this.

We have expanded the discussion section (‘Rationale for the three component scores’) to address these comments. 

We argue that not only that the value of metrics are inherently subjective, but that this subjective aspect is frequently glossed over when evaluating impact (for example, no rationale is provided for the weightings in the Altmetric Attention Score). We provide and framework that transparently incorporates subjective values with quantitative analyses to provide a useful tool for analysis.

The EMPIRE Index itself is, therefore, only directly relevant to stakeholders who share broadly similar perspectives to our panel. However, we believe the process used for developing the EMPIRE Index is of wider interest. Any interested party can reweight the EMPIRE index using subjective values arrived at using their preferred method. 

We have added these considerations to the conclusion.

We agree that there is a large literature on the use and misuse of altmetrics and had briefly summarize some key considerations into the original submission. We have expanded significantly on these in the current resubmission (in the introduction and in the discussion section “Rationale for the three component scores’), while noting that the intention of the EMPIRE Index is not to provide a tool for evaluating the overall impact (or value) of research, but for the very much more narrowly defined question of evaluating the impact of individual publications. 

Reviewer #3

The EMPIRE Index is intended to provide a measure of the impact of a publication (rather than the full impact of a research project). We use altmetrics because they are available for any publication, which enables comparative assessment. However, we acknowledge the reviewer’s concerns and have provided responses below and in the publication. 

Note that the index provides three component scores (Social, Scholarly, and Societal). The fourth is a predictor score that has no theoretical foundation in its own right except as its potential to predict long-term EMPIRE Index scores. 

We acknowledge and agree with the summary and concerns laid out here. We do not intend that this methodological paper itself to answer them. In fact, we do not think there is a universal, objective answer. 

Rather we intend that this paper can contribute to the debate by providing a framework to interpret metrics, and by acknowledging that, while metrics have a statistical relationship, the value of metrics to a given individual will depend on the question they are trying to answer, and therefore will be inherently subjective.

By combining statistical analyses with subjective insights, we present one potential approach to addressing this problem in a limited context (biomedical research publications). We have added a note to this effect in the conclusions. 

The qualitative understanding of metrics is, we agree, very important to understanding publication impact. This problem can be addressed through natural language processing and other techniques that are outside the scope of the current publication. However, we feel that purely quantitative metrics are useful in their own right.

We have noted the Becker model in the introduction and clarified the difference in scope and intent. The Becker model is conceptually different to the EMPIRE Index in several ways. 

• The Becker model is much broader in scope, as it assesses the full impact of a research project, rather than the impact of publications. The EMPIRE Index could therefore be considered to address a subset of the Becker Model. 

• The Becker model is a list of indicators, but it does not address how to synthesise them to enable comparisons. It is not quantitative.

In addition, there are practical differences in the approach. The EMPIRE Index is restricted to metrics that allow direct comparisons. For example, among the publication metrics listed in the Becker model is downloads. We excluded downloads from the EMPIRE Index because they are not available for many publications. Furthermore, the EMPIRE Index uses only metrics that can be collected automatically.

We agree that statistical assessment of sparse indicators is problematic. Wikipedia mentions are likewise sparse. We found negligible correlations between these indicators and the other metrics, likely because of these zero inflated distributions. also agree that, as with other metrics, there are likely to be interactions. 

However, citations in these documents were considered by our panel to be very important and distinct from other types of citations. Therefore, we feel it is important to include these impact measures as part of an overall assessment of publication impact. We view metrics as a starting point for qualitative examination – for example, if it is identified that a publication has had high societal impact, that should be a trigger to investigate further to understand the nature of that impact (which guidelines, in which context etc). 

We agree that some aspects of these data are problematic – in fact all altmetrics measures can be criticised for being incomplete or inaccurate. News articles depend on somewhat arbitrary lists that are biased to English language, and the lists can change over time. Guidelines in PlumX currently only include those listed in PubMed; as a result, many are missing. Policy documents re similarly skewed (our observations are that for biomedical science they are primarily sourced from NICE and IQWiG), and all of these data are subject to change as new sources are included or excluded from monitoring.

Despite these problems with the underlying data, we believe that altmetrics including policy and guidelines mentions are worthy of serious attention. We feel that this is supported by our observation of the relatively high Societal impact of the ‘notable publications’ selected by NEJM editors. 

We have added more details of the group discussion to the methods and results section.

We have changed the presentation of figure 2 to address this. 

We agree with the reviewer concerns here and note that for these reasons we do not include citescore as an impact measure in the EMPIRE Index. 

However, for the predictor scores we are trying to estimate from early data what later impact might be. The choice of journal can have a significant effect on the impact of the publication, primarily because of readership (i.e. some journals have significantly higher reach into key audiences). Unfortunately, there is no consistent, publicly available measure of journal reach measure, because most publishers don’t make readership figures available in a comparable format. We therefore included Citescore not as a measure of impact, but as a proxy for journal reach. 

This is mentioned in the methodology but we have strengthened this in the discussion section (‘Predictor scores’) . 

We have included two case studies to illustrate the potential use of the index, as well as referenced two studies that we have conducted using the EMPIRE Index (both have been presented at conferences but not yet published).

We agree and feel that this has been addressed in the discussion already. However, we acknowledge there is much more that can be said on this topic, and have strengthened the discussion section accordingly (‘Rationale for the three component scores’). 

We acknowledge the reviewer’s concerns. The EMPIRE Index was developed to address an identified need within Novartis and the community of medical publications professionals that support the communication of research sponsored by the pharmaceutical industry. The methodology and results have been presented to meetings of the International Society for Medical Publication Professionals on three occasions, as an oral and two poster sessions). These presentations are available on the Figshare project page: https://figshare.com/projects/EMPIRE_EMpirical_Publication_Impact_and_Reach_Evaluation_Index/85211

We feel that the approach taken and the insights gained from the development of the EMPIRE index may be of interest to others involved in the communication of scientific research. Our intention in seeking publication is not to present a definitive answer to the problem of research evaluation, but rather to share what we have learned and contribute to the debate.

---

## [Decision Letter · Decision Letter 1]

2 Mar 2022

Introducing the EMPIRE Index: A novel, value-based metric framework to measure the impact of medical publications

PONE-D-21-11452R1

Dear Dr. Tomas James Rees,

We’re pleased to inform you that your manuscript has been judged scientifically suitable for publication and will be formally accepted for publication once it meets all outstanding technical requirements.

Kind regards,

Leila Nemati Anaraki

Guest Editor

PLOS ONE

Additional Editor Comments (optional):

Reviewers' comments:

Reviewer's Responses to Questions

**Comments to the Author**

1. If the authors have adequately addressed your comments raised in a previous round of review and you feel that this manuscript is now acceptable for publication, you may indicate that here to bypass the “Comments to the Author” section, enter your conflict of interest statement in the “Confidential to Editor” section, and submit your "Accept" recommendation.

Reviewer #1: All comments have been addressed

Reviewer #2: All comments have been addressed

Reviewer #3: All comments have been addressed

2. Is the manuscript technically sound, and do the data support the conclusions?

Reviewer #1: Yes

Reviewer #2: Yes

Reviewer #3: Yes

3. Has the statistical analysis been performed appropriately and rigorously? 

Reviewer #1: Yes

Reviewer #2: Yes

Reviewer #3: Yes

4. Have the authors made all data underlying the findings in their manuscript fully available?

Reviewer #1: Yes

Reviewer #2: Yes

Reviewer #3: Yes

5. Is the manuscript presented in an intelligible fashion and written in standard English?

Reviewer #1: Yes

Reviewer #2: Yes

Reviewer #3: Yes

6. Review Comments to the Author

Reviewer #1: Hi Dear Sir/Madam,

The article is now appropriate to publish in this journal. all the review recommendations have revised.

Sincerely Yours,

Reviewer #2: (No Response)

Reviewer #3: Thanks to the authors for their thorough response. I feel much better about the paper with the added information and context. The case uses and recommendation for use really helps contextualize the metric, to (hopefully) prevent misuse and abuse of this metric, since it is highly tailored to a specific audience and purpose, and the added details about the panel help to better understand the viewpoints of the stakeholders. I still think the metric has too much subjectivity for broader implementation, but feel that the manuscript adequately addresses this concern, and I appreciate the care that was taken to contextualize the decisions that were made.

7. PLOS authors have the option to publish the peer review history of their article (what does this mean?). If published, this will include your full peer review and any attached files.

Reviewer #1: No

Reviewer #2: No

Reviewer #3: No

---

## [Editor Report · Acceptance letter]

4 Mar 2022

PONE-D-21-11452R1 

Introducing the EMPIRE Index: A novel, value-based metric framework to measure the impact of medical publications 

Dear Dr. Rees:

I'm pleased to inform you that your manuscript has been deemed suitable for publication in PLOS ONE. Congratulations! Your manuscript is now with our production department. 

Kind regards, 

on behalf of

Dr. Leila Nemati Anaraki 

Guest Editor

PLOS ONE